# Intranasal trimeric sherpabody inhibits SARS-CoV-2 including recent immunoevasive Omicron subvariants

Anna R. Mäkelä[1], Hasan Uğurlu [1], Liina Hannula[2], Ravi Kant [1,3], Petja Salminen[1], Riku Fagerlund[1], Sanna Mäki[1], Anu Haveri [4], Tomas Strandin[1], Lauri Kareinen[1,3], Jussi Hepojoki [1], Suvi Kuivanen[1], Lev Levanov[1], Arja Pasternack [5], Rauno A. Naves[5], Olli Ritvos[5], Pamela Österlund [4], Tarja Sironen [1,3], Olli Vapalahti [1,3,6], Anja Kipar [3,7], Juha T. Huiskonen [2], Ilona Rissanen [2] & Kalle Saksela [1,6] ✉

The emergence of increasingly immunoevasive SARS-CoV-2 variants emphasizes the need for prophylactic strategies to complement vaccination in fighting the COVID-19 pandemic. Intranasal administration of neutralizing antibodies has shown encouraging protective potential but there remains a need for SARS-CoV-2 blocking agents that are less vulnerable to mutational viral variation and more economical to produce in large scale. Here we describe TriSb92, a highly manufacturable and stable trimeric antibody-mimetic sherpabody targeted against a conserved region of the viral spike glycoprotein. TriSb92 potently neutralizes SARS-CoV-2, including the latest Omicron variants like BF.7, XBB, and BQ.1.1. In female Balb/c mice intranasal administration of just 5 or 50 micrograms of TriSb92 as early as 8 h before but also 4 h after SARS-CoV-2 challenge can protect from infection. Cryo-EM and biochemical studies reveal triggering of a conformational shift in the spike trimer as the inhibitory mechanism of TriSb92. The potency and robust biochemical properties of TriSb92 together with its resistance against viral sequence evolution suggest that TriSb92 could be useful as a nasal spray for protecting susceptible individuals from SARS-CoV-2 infection.

The success of the current vaccines in the fight against the SARS-CoV-2 pandemic is challenged by the emergence of viral variants of concern (VOCs) that show strong resistance to neutralizing antibodies induced by vaccinations or prior infection. Moreover, immune disorders or other health conditions can preclude appropriate vaccine responses in many individuals. In addition to pharmaceuticals for treating COVID-19 disease, new approaches for preventing transmission and spreading of SARS-CoV-2 are therefore urgently needed.

Since the nasal epithelium of the respiratory tract is the first dominant replication site of SARS-CoV-2 preceding virus transport into the lung[1], intranasal administration of SARS-CoV-2 neutralizing agents poses an attractive prophylactic concept. In animal models,

[1]Department of Virology, University of Helsinki, Helsinki, Finland. [2]Institute of Biotechnology, Helsinki Institute of Life Science HiLIFE, University of Helsinki, Helsinki, Finland. [3]Department of Basic Veterinary Sciences, University of Helsinki, Helsinki, Finland. [4]Finnish Institute for Health and Welfare, Helsinki, Finland. [5]Department of Physiology, University of Helsinki, Helsinki, Finland. [6]HUS Diagnostic Centre, HUSLAB, Clinical Microbiology, Helsinki University Hospital, Helsinki, Finland. [7]Laboratory for Animal Model Pathology, Institute of Veterinary Pathology, Vetsuisse Faculty, University of Zurich, Zurich, Switzerland. ✉e-mail: kalle.saksela@helsinki.fi

monoclonal antibodies targeted against the spike envelope protein of SARS-CoV-2 have shown to be effective for COVID-19 prophylaxis[2–6]. To avoid mutational escape from neutralization, such antibodies have been used as cocktails and targeted against conserved regions of spike[7]. Markedly increased potency could also be achieved by constructing a pentameric IgM version of spike-targeted IgG antibodies[4].

To develop more economical and manufacturable anti-SARS-CoV-2 reagents, many laboratories have moved from monoclonal antibodies to smaller antibody fragments, including single variable domain-based nanobodies. Several of these have shown promising antiviral efficacy in cell culture or animal models, including utility as an intranasal SARS-CoV-2 prophylaxis[8–11].

Compared to these, small antibody-mimetic scaffold proteins are even more robust, versatile to engineer, and cheaper to produce for diverse biological targeting purposes[12]. Sherpabodies (SH3 Recombinant Protein Affinity) are very small (< 60 aa) targeting modules that constitute a recent addition to the toolbox of scaffold targeting proteins and comprise of an improved SH3-domain fold derived from the human ciliary adaptor protein nephrocystin[13]. Here, we report engineering of a sherpabody targeted against a highly conserved region in the spike receptor binding domain (RBD) into a potent prophylactic anti-SARS-CoV-2 agent.

## Results

### Development of sherpabody-based SARS-CoV-2 inhibitors

With the aim of developing inhibitory proteins that could be used for neutralizing SARS-CoV-2, we screened a large antibody-mimetic phage library displaying ~$10^{11}$ unique sherpabodies (Fig. 1a, b) by using the receptor binding domain (RBD) of the spike glycoprotein of the Wuhan-Hu-1 strain as an affinity bait. Starting from the second round of affinity selection and phage amplification (panning), a clear enrichment of RBD-binding clones was observed. After three panning cycles, 192 individual sherpabody-displaying phage clones were isolated and found to represent 15 different sequences. These 15 unique sherpabodies were tested in phage-ELISA for binding to RBD-mFc, control mFc, or a monoclonal antibody against the E-tag peptide used for monitoring sherpabody display efficiency (see Fig. 1b). All 15 sherpabody clones mediated strong and apparently specific RBD binding (Fig. 1c). Clone number 92 (Sb92; amino acid sequence: EEYIAVGDF FSTDPADLTFKKGEILLVIERGTSAGDGWWIAKDAKGNEGLVPRTYLEPYS) was among the strongest RBD binders and was chosen for further development due to its ability to also bind to RBD of SARS-CoV-1 (see later), suggesting a conserved target epitope that would be unlikely to vary between different variants of SARS-CoV-2. Sb92 was produced as a GST-fusion protein in bacteria and its RBD-binding affinity was evaluated using a semi-quantitative antigen capture-ELISA (Fig. 1d), which indicated an affinity ($K_D$) of 30 nM.

### Neutralization of SARS-CoV-2 and SARS-CoV-1 by trimerized Sb92

To enhance targeting of the trimeric cell surface spike protein we constructed a multimerized derivative of Sb92 (TriSb92) containing three tandem copies of Sb92 connected by flexible 15-mer Gly-Ser linkers. To evaluate the potency of TriSb92 as a neutralizing agent we incubated different viral isolates with serial dilutions of it before infection of VeroE6-TMPRSS2-H10 cultures. When the original Wuhan strain of SARS-CoV-2 was tested a dose-dependent decrease in the number of infected cells was observed corresponding to an impressive $IC_{50}$ value of 0.24 nM (Fig. 2). When clinical SARS-CoV-2 isolates representing the Beta (B.1.351), Delta (B.1.617.2), and Omicron (B.1.1.529) BA.1 variants of concern were tested we observed an even more potent neutralization by TriSb92, resulting in $IC_{50}$ values of 0.11 nM (Beta), 0.18 nM (Delta), and 0.08 nM (BA.1). Finally, we tested also a more recent viral isolate corresponding to the Omicron BA.5 variant that carries several additional mutations in the RBD of its spike protein that

a

RT-loop

```
EEYIAVGDFXXXXXXDLTFKKGEILLVIEX
XXXXXDGWWIAKDAKGNEGLVPRTYLEPYS
```

n-Src-loop

b

Phage coat protein pIII

E-tag

c

| Plate coating | #51 | #52 | #53 | #54 | #55 | #56 | #57 | #58 | #59 | #60 | #61 | #62 | #92 | #102 | #103 |
|---|---|---|---|---|---|---|---|---|---|---|---|---|---|---|---|
| | | | | | | | Phage clone number | | | | | | | | |
| RBD-mFc | | | | | | | | | | | | | | | |
| mFc | | | | | | | | | | | | | | | |
| anti-E-tag | | | | | | | | | | | | | | | |

Phage ELISA signal:  Undetectable ▬ High

d

Sb92 coating (µg/ml): 1.5, 1.0, 0.67, 0.44

**Fig. 1 | Discovery of RBD-targeting sherpabodies. a, b** Six and three residues in the RT- and n-Src-loops, respectively, of the human nephrocystin SH3 domain were replaced with hexapeptides (in red) comprising random combinations of the 20 natural amino acids to create a large semisynthetic M13 phage library displaying ~$10^{11}$ individual sherpabodies with unique binding surfaces formed by the 12 randomized loop residues shown red in the illustration based on the structure pdb 1S1N (https://www.ncbi.nlm.nih.gov/Structure/pdb/1S1N)[13]. **c** Heat map showing relative binding in an ELISA assay of fifteen affinity-selected sherpabody-displaying phage clones targeted to RBD of SARS-CoV-2 spike protein (RBD-mFc), negative control (mFc), or an antibody evenly recognizing all displayed clones (anti-E-tag). **d** Semi-quantitative ELISA-based analysis of the RBD-binding affinity (30 nM) of the sherpabody clone Sb92. The experiments were repeated independently three times with similar results. A representative assay is shown. Source data are provided as a Source Data file.

provide it with a problematic capacity for evading neutralizing antibody responses. We noted that despite being slightly more resistant than the older variants tested, Omicron BA.5 could also be efficiently neutralized by TriSb92 with a subnanomolar $IC_{50}$ value of 0.82 nM.

To extend these studies to a wider panel of viruses, we utilized a pseudovirus model based on luciferase expressing lentiviral vectors infecting ACE2-expressing HEK293T cells in a spike-dependent manner[14]. Testing of Wuhan, Beta, Delta, Omicron BA.1 and BA.5 spike proteins in this model showed that pseudoviruses are slightly more sensitive to neutralization by TriSb92, but gave results that correlated well with our data on the corresponding authentic SARS-CoV-2 strains (Fig. 2). The $IC_{50}$ values derived from these experiments (0.06 nM for Wuhan, 0.03 nM for Beta, 0.06 nM for Delta, 0.11 nM for Omicron BA.1, and 0.35 nM for Omicron BA.5) thus both support our data from using live viruses and help to validate this pseudovirus model for studying the inhibitory potency of TriSb92 against additional virus strains.

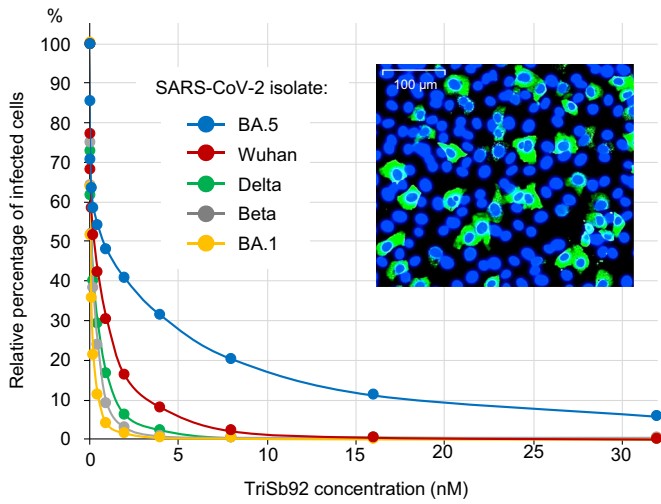

| IC$_{50}$ (nM) | | |
|---|---|---|
| *Authentic viruses* | | |
| Wuhan | 0.24 | ±0.06 |
| Beta | 0.11 | ±0.04 |
| Delta | 0.18 | ±0.05 |
| BA.1 | 0.08 | ±0.01 |
| BA.5 | 0.83 | ±0.29 |
| *Pseudoviruses* | | |
| Wuhan | 0.06 | ±0.02 |
| Beta | 0.03 | ±0.01 |
| Delta | 0.06 | ±0.04 |
| BA.1 | 0.11 | ±0.03 |
| BA.3 | 0.03 | ±0.01 |
| BA.2 | 0.25 | ±0.04 |
| BA.2.13 | 0.19 | ±0.10 |
| BA.4/5 | 0.35 | ±0.04 |
| BA.2.3.20 | 0.08 | ±0.01 |
| BF.7 | 0.18 | ±0.03 |
| BQ.1.1 | 1.74 | ±0.30 |
| XBB | 1.19 | ±0.49 |
| BU.1 | 0.82 | ±0.03 |
| SARS-CoV-1 | 5.65 | ±0.25 |

**Fig. 2 | Neutralization of SARS-CoV-2 and SARS-CoV-1 by TriSb92.** Half-maximal inhibitory concentration (IC$_{50}$) of TriSb92 against different authentic SARS-CoV-2 variants or pseudoviruses using the indicated spike proteins were established based on automated counting of fluorescently labelled infected cells (see figure insert) or luciferase reporter gene expression, respectively. Standard deviation for the calculated IC$_{50}$ values is shown. The pseudovirus neutralization experiments were repeated independently three or more times with similar results. Average and standard deviations of representative assays performed in duplicates is shown. Neutralization of authentic viruses were repeated with similar results three times for Wuhan, twice for Delta, BA.1, and BA.5 and once for Beta in experiments performed in quintuplicate and involved counting of approximately 25,000 cells per condition after immunostaining illustrated in the figure insert. The original data are provided as a Source Data file. An alignment of RBD sequences of spike variants used in this study is shown in Fig. S1.

Recently several BA.5 subvariants with even higher immunoevasive capacity and ability to escape from neutralization by the currently available therapeutic antibodies have been reported to gain prevalence[15]. To test the potency of TriSb92 against such challenging SARS-CoV-2 strains we generated pseudoviruses carrying spike proteins from a panel of viruses that have emerged and caused concern around the world in mid- and late-2022 (Fig. 2 and Fig. S1). All these pseudoviruses could also be efficiently neutralized by TriSb92, although the IC$_{50}$ values had increased from the subnanomolar range to 1–2 nM. Finally, we created a pseudovirus based on the spike glycoprotein of SARS-CoV-1, and found that it could also be readily neutralized by TriSb92, albeit with a somewhat lower efficiency (IC$_{50}$ 5.65 nM) than the SARS-CoV-2 variants tested (Fig. 2).

Together, these data show that TriSb92 is a potent inhibitor of SARS-CoV-2 infection, which targets a conserved site in the RBD of spike in a manner that is not prevented by mutations found in the older as well as the currently prevalent and emerging SARS-CoV-2 variants, and provides TriSb92 with a capacity to neutralize a related sarbecovirus (SARS-CoV-1).

### TriSb92 induces a conformational change in the S2 subunit of the spike protein

To understand the mechanism behind the potent SARS-CoV-2 neutralization by TriSb92 we examined whether it inhibited attachment of spike-dependent pseudovirus particles to target cells. To this end, these virions were either untreated or treated for 30 min with 100 ng/ml of TriSb92 (a concentration sufficient to fully suppress infectivity) or a high-titre COVID-19 convalescent serum before incubation with ACE2-expressing HEK293 cells, followed by washing of unbound virions and Western blot analysis of lysates of these cells for the p24 pseudoviral capsid protein. We found that treatment with TriSb92 did not affect target cell binding of the spike-pseudotyped virions, whereas this was completely blocked by the convalescent serum used as a control (Fig. 3a).

While this result ruled out displacement and shedding of the S1 subunit from the spike complex as the principal mechanism of

action of TriSb92, it was still of interest to compare the relative amounts of S1 and S2 subunits and uncleaved spike (S0) protein present in TriSb92-treated and untreated virions. Concentrated virus preparations were lysed in SDS-PAGE loading buffer and either warmed to 37 °C or boiled under reducing conditions to help detecting different types of epitopes in Western blots, which were probed with polyclonal rabbit sera specific for S1 or S2, or with the human convalescent serum (Fig. 3c–e).

Both the rabbit anti-S1 and the human anti-spike sera were able to detect S1 only if the samples had not been boiled. Use of these antibodies clearly showed that the amount of virion-associated S1 was not affected by TriSb92 treatment, thus further ruling out S1 shedding as the inhibitory mechanism of TriSb92. The rabbit anti-S2 and the human anti-spike sera readily detected S2 protein irrespective of the preheating temperature, but stronger S2 signals were obtained after boiling. In the case of the boiled samples TriSb92-treatment had no effect on the amount of S2 detected by these antisera, as one would expect for a membrane-anchored viral protein. However, whereas S2 was detected by both antisera in the unboiled control sample, it could not be seen in the corresponding TriSb92-treated sample. A similarly striking behaviour was observed for the intact spike protein (S0) where the S2 subunit remains covalently connected to S1, as the rabbit anti-S1 and the human anti-spike sera could detect S0 only in the unboiled control but not in the TriSb92-treated sample.

Based on these observations we conclude that regardless of the proteolytic cleavage status of the spike protein binding of TriSb92 to the RBD of its S1 subunit induces a conformational change in the S2 subunit, which depending on the experimental condition can affect detection of cleaved S2 and the uncleaved S0 precursor in Western blotting.

### Intranasally administered TriSb92 provides efficient prophylactic pre- and post-exposure protection in mouse models of SARS-CoV-2 infection and disease

To evaluate the prophylactic efficiency of TriSb92 in vivo, we used a recently described animal model of COVID-19 where Balb/c mice are

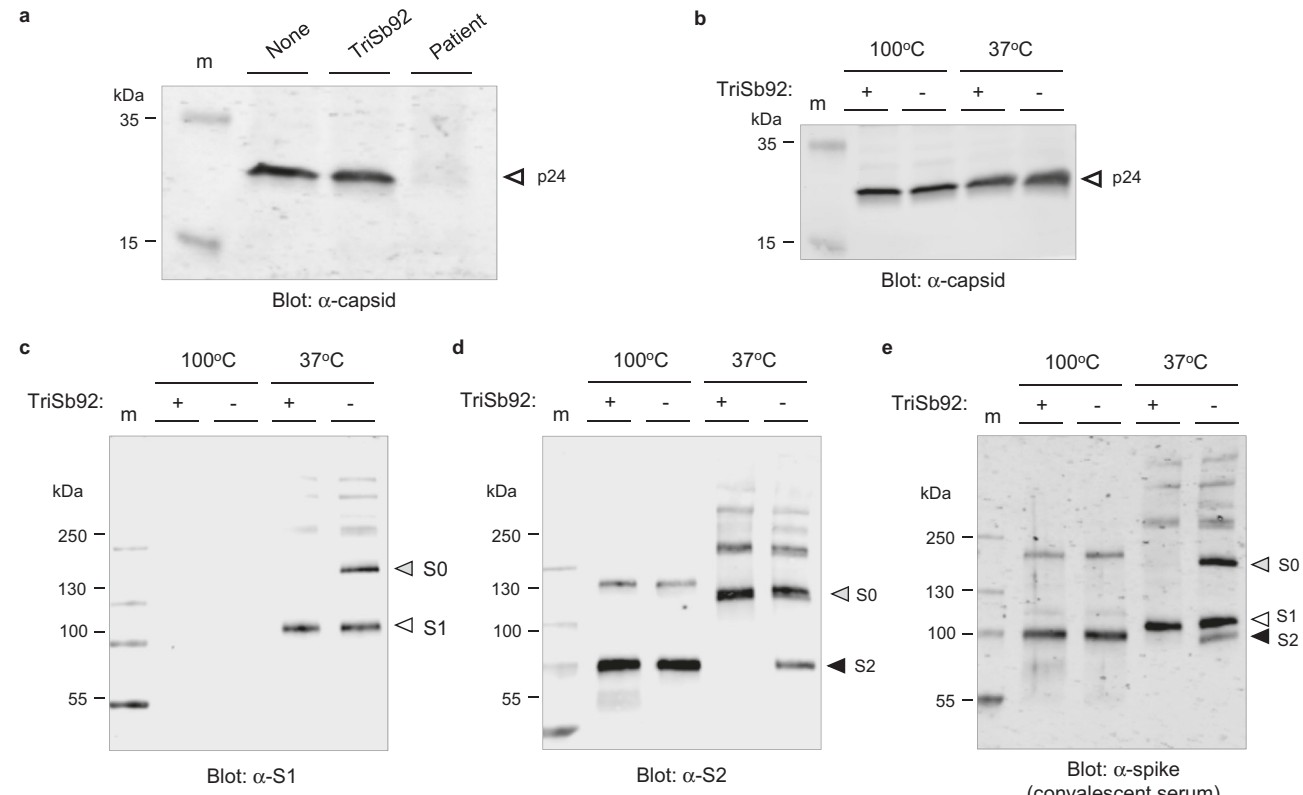

**Fig. 3 | TriSb92 does not prevent virus attachment to target cells but prevents infection by inducing a conformational change in spike. a** Spike-decorated pseudoviruses were treated with TriSb92, COVID-19 convalescent serum (Patient), or left untreated (None) before incubation with ACE2-expressing HEK293 cells. The amount of cell-associated pseudoviral capsid protein (p24) after washing of the cultures was determined by Western blotting. **b**–**e** Spike-pseudotyped virions treated (+) or not treated (−) with TriSb92 were lysed in SDS-PAGE loading buffer and warmed (37 °C) or boiled (100 °C) before analysis by Western blotting using antibodies specific for the p24 capsid protein (**b**), spike S1 subunit (**c**), spike S2 subunit (**d**), or COVID-19 convalescent serum that can recognize S1 and S2 (**e**). The experiment 3a-e was repeated independently three times with similar results. Uncropped original scans of all blots presented in panels **a**–**e** are provided as Source Data files.

intranasally inoculated with the SARS-CoV-2 Beta (B.1.351) variant[16]. At different time points before the SARS-CoV-2 challenge mice received intranasal administration of TriSb92. In the first experiment, mice received 25 µg of TriSb92 per nostril and were challenged intranasally one hour later with $2 \times 10^5$ PFU of SARS-CoV-2 B.1.351. This amount of TriSb92 was chosen to represent a relatively modest dose, which is in the range also referred as ultra-low in the SARS-CoV-2 antibody inhibitor literature[5].

Animals were euthanized at 2 days post infection (dpi) and the right lungs subjected to quantitative real-time PCR (RT-qPCR) for viral RNA. In untreated control mice, the lungs exhibited abundant SARS-CoV-2 subgenomic E RNA (median Cq value 18.45), whereas none could be detected in the lungs of the mice that had received a prophylactic dose of TriSb92 (Fig. 4a). The results were confirmed by histology and immunohistology for the detection of SARS-CoV-2 nucleoprotein (NP) in nasal mucosa, airways, and left lungs (Fig. 4b). In control animals, viral antigen was present throughout the respiratory tract, in the nasal cavity (respiratory and olfactory epithelial cells), in bronchiolar epithelial cells and in pneumocytes in adjacent alveoli in association with a mild acute bronchointerstitial pneumonia. Viral antigen was also detected within individual macrophages in bronchiolar lymph nodes. In contrast, treated animals were entirely free of viral antigen in nose, lower airways, lungs, and bronchiolar lymph nodes and did not exhibit any pathological changes (Fig. 4b).

In a second larger experiment, the protective effect of the previously used dose of TriSb92 was compared with a 10-fold lower dose (2.5 µg per nostril) similarly administered 1 h before SARS-CoV-2 challenge. In addition, we tested the effect of the 25 µg per nostril administered 4 h or 8 h before virus inoculation. As shown in Fig. 4a, also the lower dose of TriSb92 efficiently protected the animals from subsequent SARS-CoV-2 challenge 1 h later (no detectable sgE RNA in lungs 3 days postinfection); however, in two of these animals viral antigen was detected in a few patches of bronchiolar epithelial cells. TriSb92 was very effective also when given 4 h before the viral challenge. None of these animals showed evidence of viral antigen expression in the lung, and in only one of the four animals low but detectable levels of sgE RNA were found. Likewise, when TriSb92 was given 8 h before the challenge the lungs of 3 of the 4 mice were clean of sgE RNA and viral antigen, whereas one animal did show substantial albeit reduced viral RNA levels and also expressed viral antigen within bronchiolar epithelial cells.

A third experiment was carried out to examine the early therapeutic potency of TriSb92 when administered after SARS-CoV-2 infection of epithelial cells had already occurred. The same dosing of 25 µg of TriSb92 per nostril and a challenge with $2 \times 10^5$ PFU of SARS-CoV-2 B.1.351 were again used. The results on the five unprotected animals recapitulated our earlier data, showing abundant sgE RNA in the lungs together with widespread viral antigen expression in the bronchiolar and alveolar epithelium (Table S2), whereas no sgE RNA or viral antigen could be detected in any of the five control mice that received a prophylactic TriSb92 treatment 2 h before the challenge.

Strikingly, close to complete lack of sgE RNA in the lungs was also observed at 2 days after the high-dose SARS-CoV-2 challenge when TriSb92 was administered 2 h or even 4 h after the infection (Fig. 4a). No Cq value for sgE RNA could be obtained for 4 of the 5 animals treated 4 h post-infection, while Cq 32,66 was measured for one animal

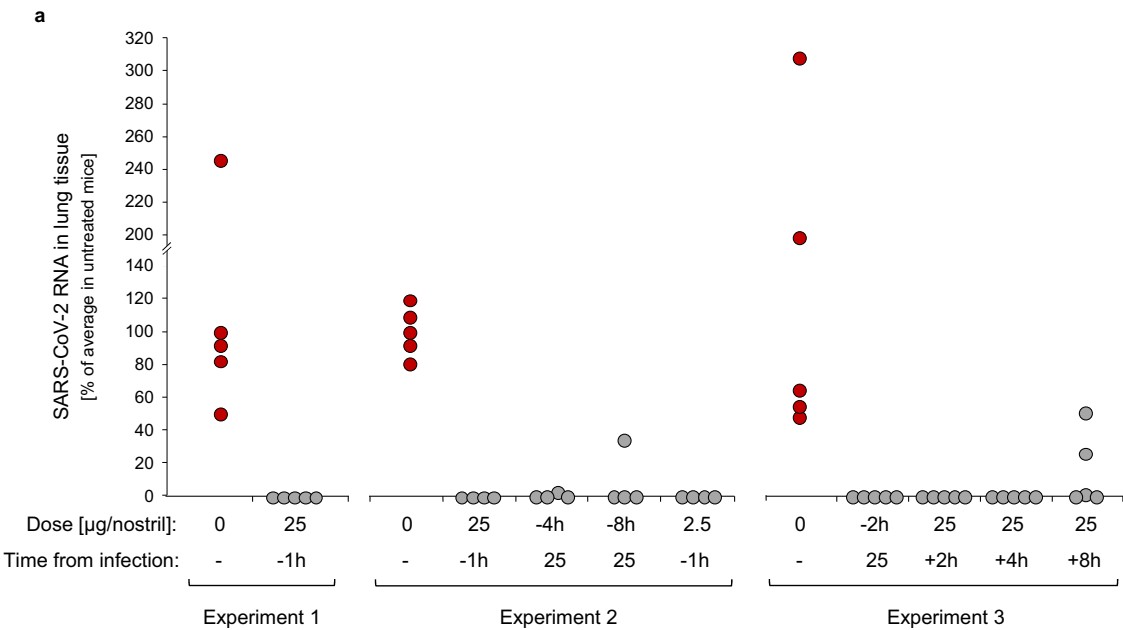

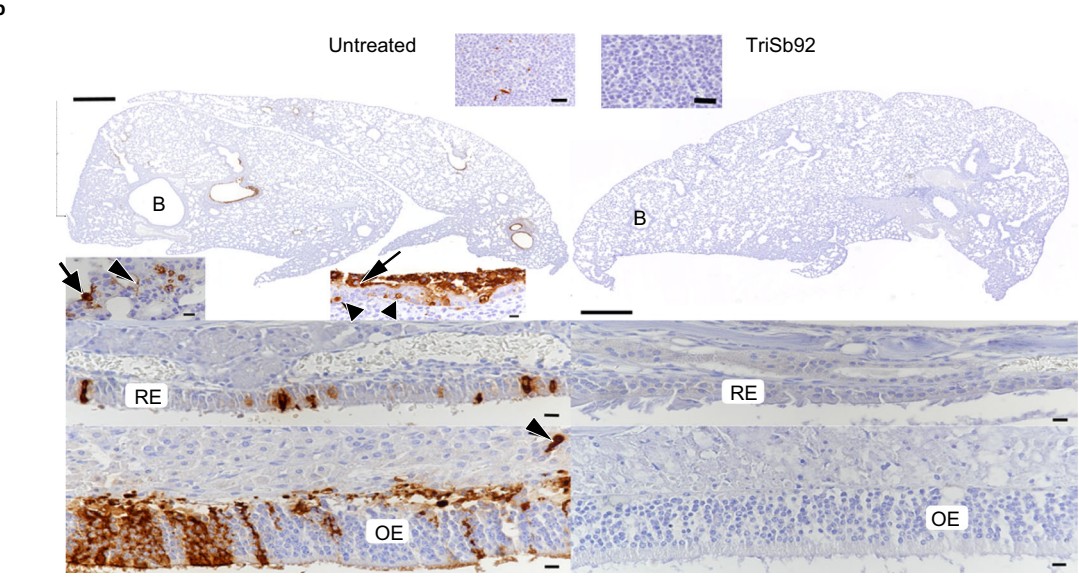

**Fig. 4 | Prophylactic and early therapeutic efficacy of TriSb92 in mice.**
**a** Subgenomic viral RNA was quantified in lung tissues of Balb/c mice 2 days after challenge with $2 \times 10^5$ PFU of SARS-CoV-2 B.1.351 in three independent infection experiments. The infected mice were treated with intranasal administration TriSb92 at the indicated doses and times before or after the infection. **b** SARS-CoV-2 nucleoprotein immunostaining in the nasal cavity and lung tissues 2 days after infection in an untreated mouse (left) and in a mouse pretreated with 25 μg of TriSb92 per nostril (right). The untreated animal exhibits viral antigen expression in the nasal respiratory epithelium (RE) and olfactory epithelium (OE). In the lung, several bronchioles (B) exhibit viral antigen in a variable number of epithelial cells that are often degenerate (arrowheads) or sloughed off (arrows). In infected alveoli, both type I pneumocytes (arrowhead) and type II pneumocytes (arrow) are positive. The bronchial lymph node (top inset) exhibits several positive cells (macrophages or dendritic cells). The animal that received TriSb92 does not exhibit any evidence of viral antigen expression in nasal epithelium, lung and bronchial lymph node. Bars: lung overviews – 500 μm; bronchial lymph nodes – 20 μm; all others – 10 μm. **c** The three mouse infection experiments shown involving 60 animals were each performed once. Original data on RNA quantification are provided as Source Data files. Panel **b** shows representative immunostaining findings, and a detailed analysis of all animals are described in Table S2.

(indicating sgE RNA levels < 0.01% of the average in the five untreated mice). TriSb92 given 8 h after SARS-CoV-2 was still very beneficial, although an efficient suppression of viral sgE RNA expression in the lungs could be seen only in 3/5 of the mice (undetectable in two, 1.3% of untreated in one). Perhaps due to the initial establishment of the infection, however, the mice treated with TriSb92 after the infection were not equally devoid of viral antigen in their lungs compared to the mice that received TriSb92 prophylactically. SARS-CoV-2 nucleoprotein could be detected in the same cell types, although being markedly

less widespread and extensive than in the untreated animals (see Table S2). Moreover, whereas in the untreated mice the infection of the bronchiolar epithelium was associated with degenerative changes, no such pathological alterations were observed in mice treated post-infection with TriSb92.

To extend these studies to an animal model involving more severe disease pathogenesis we used the mouse-adapted SARS-CoV-2 virus maVie16 that induces profound disease and mortality in Balb/c mice recapitulating many aspects of human COVID-19[17]. Also of note,

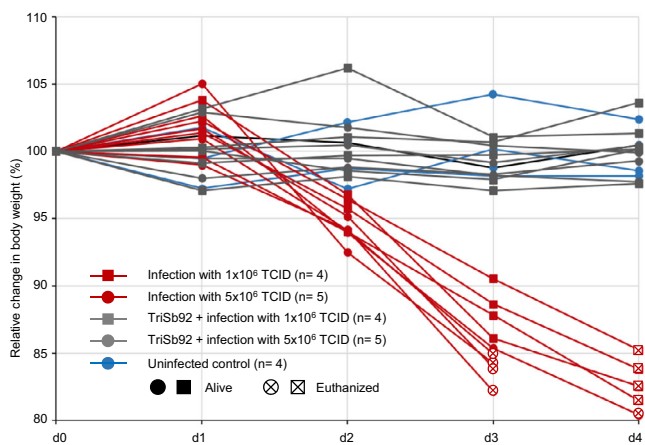

**Fig. 5 | TriSb92 can protect from a lethal challenge with mouse-adapted SARS-CoV-2.** Body weight development of Balb/c mice that were infected with $1 \times 10^6$ or $5 \times 10^6$ PFU of the pathogenic SARS-CoV-2 strain maVie16 following intranasal administration TriSb92 (25 µg of per nostril; red lines and symbols) or the same volume of saline buffer (grey lines and symbols) 1 h earlier, as indicated. Four control mice received similar mock treatment but no TriSb92 or virus (blue lines and symbols). The experiment was terminated at day 4 post-infection when all infected mice that had not received TriSb92 prophylaxis had been euthanized because of losing 15% or more of their weight (crossed empty symbols). This experiment was performed once. Original data on mouse weight development are provided as Source Data files.

in resemblance to the recently emerged immunoevasive Omicron variant derivatives (such as BJ.1, BA.2.82, BH.1) the spike protein of maVie16 has acquired mutations affecting the RBD residues K417, Q493, and Q498. We used a lower ($1 \times 10^6$ TCID) or a higher ($5 \times 10^6$ TCID) dose of maVie16 to infect two groups of nine Balb/c mice, half of which had received TriSb92 intranasally (25 µg/per nostril) one hour earlier, and then monitored possible disease progression by daily body weight recording (Fig. 5). Four mice that received just mock treatment but no TriSb92 or virus were included as controls and likewise monitored. All nine infected animals that did not receive TriSb92 experienced significant weight loss at day 2, which at day 3 was so severe ($\geq 15\%$) that 3 of 4 mice in the higher viral dose group and at day 4 all of the remaining 6 control mice had to be euthanized. By contrast, none of the mice that received a prophylactic dose of TriSb92 showed any signs of weight loss (Fig. 5) or could be distinguished from the uninfected control animal based on their apparent well-being or behaviour.

Together, these data show that in these two animal models intranasal administration of TriSb92 at relatively low doses can provide an impressive protection and postexposure prophylactic effect against SARS-CoV-2 infection as measured by virological and histopathological parameters as well as objective severity of disease progression.

## Feasibility of TriSb92 for use as a prophylactic formulation

To assess the feasibility of formulating TriSb92 into an intranasal spray or drop for human use, we examined some of its key properties in this regard, namely in vitro toxicity, potential cross-reactivity against nasal epithelium, and stability under relevant storage temperatures. The results of these studies were impressive. More specifically, even when applied at very high concentration (1000 µg/ml) to human primary endothelial cell (hNECs) culture, TriSb92 caused no signs of reduced cell viability, as analyzed by measuring cellular metabolic activity (Fig. S2). When binding of TriSb92 to extracts of hNECs was tested in an ELISA-like assay, no signal above the background of the assay was seen with any of the tested TriSb92 concentrations (as high as 10 µg/ml), whereas a strong binding signal was observed from extracts prepared from the same cells transduced to express SARS-CoV-2 RBD (Fig. S3).

Finally, we did not observe any loss of neutralizing activity of TriSb92 obtained from our research grade purification protocol and stored in PBS with no added preservatives or inhibitors up to 15 months in ultra-low freezer (−80 °C), regular freezer (−20 °C), refrigerator (+4 °C), or at room temperature (+20−25 °C) (Fig. S4).

## TriSb92 targets a conserved region in the RBD

To gain structural and mechanistic insight into the broadly cross-neutralizing action of TriSb92, cryogenic electron microscopy (cryoEM) studies were carried out. Single-particle cryoEM analysis of the prefusion SARS-CoV-2 S trimer mixed with TriSb92 revealed three particle populations distinguished by the up and down conformations characteristic of sarbecoviral RBDs[18–20], with the majority (54%) of particles presenting an all up RBD conformation, and minority populations presenting all down and mixed up and down RBD conformations (Fig. S5 and Fig. S6). Reconstructions of the all up and all down populations yielded average estimated resolutions of 2.9 Å and 3.1 Å, respectively (Fig. S5, Fig. S7, and Table S1; 6−7 Å resolution over the flexible RBD regions). Reconstruction A describes the S-trimer with all three RBDs in the up-conformation decorated with additional density corresponding to bound TriSb92 (Fig. 6a), while reconstruction B describes the S-trimer with all three RBDs in the down-conformation without any additional density indicative of a sherpabody (Fig. S6 and Fig. S7). To corroborate that the additional density associated with the up-conformation RBDs was TriSb92-derived, we generated a model of Sb92 with I-TASSER[21] and fitted it into the additional density, confirming that this density perfectly matched with the size and shape of the SH3-fold of TriSb92 (Fig. 6b, Fig. S8). The placement of RBD and Sb92 models was unambiguous at map resolution of 6.9 Å, and identified the RT-loop of Sb92 as a key mediator of Sb-RBD interactions (Fig. S8).

As illustrated in Fig. 6c, d, TriSb92 targets a region distal from the ACE2 binding site that is highly conserved among sarbecoviruses. In line with the observed cross-neutralization (Fig. 2), only a single residue within the predicted binding site of TriSb92 diverges between SARS-CoV-2 (Wuhan-Hu-1) and SARS-CoV-1. Furthermore, apart from just three of the 27 RBD amino acids changes subsequently acquired by SARS-CoV-2 (S375F, T376A and R408S, which are located at the very edge of the TriSb92 epitope), VOCs, including the most recent Omicron subvariants, do not contain mutations involving this binding surface (Fig. 6c, d). The epitope of TriSb92 is distinct from the epitopes of therapeutic monoclonal antibodies (Fig. S9) that primarily target the ACE2 binding site and remain vulnerable to amino acid changes accruing in this region. Instead, the epitope of TriSb92 partially overlaps with the 'cryptic epitope targeted by monoclonal antibody (mAb) CR3022[22] (Fig. S9) that does not compete with binding to ACE2. Indeed, the ability of ACE2 and Sb92 to simultaneously bind to the RBD in a non-competitive manner could also be experimentally verified using a sandwich-ELISA approach (Fig. S10).

## Discussion

The emergence of SARS-CoV-2 variants that compromise vaccine efficacy and are resistant to the available therapeutic monoclonal antibodies[23–28] underscores the importance to develop molecular tools that potently neutralize VOCs. In this study, we describe a promising SARS-CoV-2 inhibitor, TriSb92, based on an antibody-mimetic targeting scaffold termed the sherpabody. TriSb92 targets a highly conserved region and could neutralize all tested VOCs and even a related sarbecovirus SARS-CoV-1.

In our hands, pseudoviruses were slightly more sensitive to neutralization than the corresponding authentic SARS-CoV-2 variants, but in both cases the concentration of TriSb92 required for half-maximal inhibition of infection ($IC_{50}$) was less than 300 pM for all pre-Omicron SARS-CoV-2 variants as well as for Omicron BA.1 and BA.3, where only one amino acid change (S375F) appears within the TriSb92 epitope.

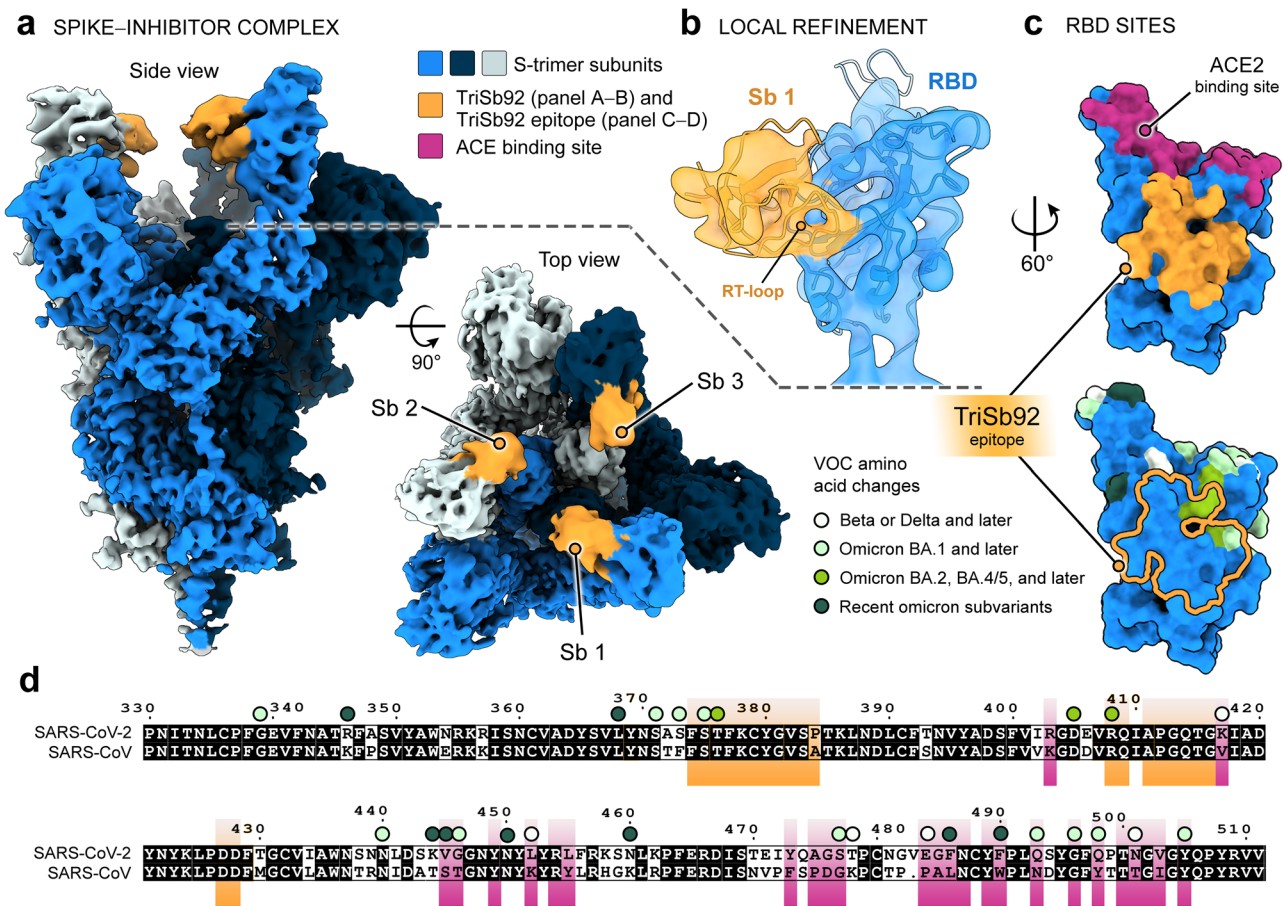

**Fig. 6 | Cryo-EM of SARS-CoV-2 S in complex with TriSb92 reveals the highly conserved TriSb92 binding site. a** Side and top views of the cryoEM reconstruction of spike trimer in complex with TriSb92. **b** Locally refined cryoEM reconstruction of the RBD region, with fitted models of Sb92 (generated with I-TASSER) and the RBD (PDB 7KMS; https://www.ncbi.nlm.nih.gov/Structure/pdb/7KMS). **c** Surface rendering of the RBD with the ACE2 binding site and the TriSb92 epitope highlighted (upper panel), and the location of amino acid changes in VOCs relative to the TriSb92 epitope highlighted (lower panel). **d** Pairwise sequence alignment of SARS-CoV-1 and SARS-CoV-2. The binding sites of TriSb92 and ACE2 are highlighted in orange and in purple, respectively. Circles above the RBD residues indicate amino acid changes in Omicron, Delta, and/or Beta variants, and are coloured as follows: white is found in Beta or Delta and later (Omicron) variants; mint green is found in Omicron BA.1 and later subvariants; medium lime green is found in Omicron BA.2, BA.4/5; dark green is found in the most recent Omicron subvariants. The mapped amino acid changes and associated variant information are presented in Fig. S1.

Evolution from BA.1 to the BA.2 and BA.5 lineages resulted in two additional RBD amino acid changes (T376A and R408S) located in the periphery of the TriSb92-binding interface, which was associated with a modest decrease in sensitivity to TriSb92. Nevertheless, BA.5 was still very effectively neutralized by TriSb92 with a subnanomolar $IC_{50}$ value measured for a clinical isolate (0.83 nM) as well as a corresponding pseudovirus (0.35 nM). Some of the more recent Omicron variants (BF.7, BA.2.13, and BA.2.3.20) were equally or even more sensitive to neutralization compared to BA.5, whereas others (BU.1, XBB, and BQ.1.1) were 2- to 5-fold less sensitive to the TriSb92. None of these newer variants contain additional amino changes in the TriSb92 binding region (see Fig. S1. Considering that the mechanism of action of TriSb92, discussed below, it could be speculated that mutations elsewhere in the S gene may have provided increased stability for the pre-fusion conformation of spike proteins encoded by some of the recent BA.5 subvariants.

In any case, all of the tested variants, including XBB, and BQ.1.1 were potently neutralized by TriSb92. This suggests that TriSb92 is likely to be effective also against future immune escape variants of SARS-CoV-2, and considering the high conservation of the TriSb92 target epitope in related animal viruses, it could also be useful as a first line of defence against pandemic human sarbecoviruses emerging in the future.

An advantage of sherpabodies over conventional antibodies and most antibody fragments is that they can be readily assembled into multimers in a context-independent manner to enhance avidity, specificity, as well as functionality. The trimeric design of TriSb92 was chosen considering the trimeric architecture of the SARS-CoV-2 spike glycoprotein. This provided TriSb92 an impressive capacity for half-maximal inhibition ($IC_{50}$) of the SARS-CoV-2 infection at picomolar concentration despite only a moderately high binding affinity ($K_D$) of 30 nM of the interaction between a monomeric Sb92 sherpabody and an individual RBD (Wuhan-Hu-1).

Besides its binding avidity, the strong neutralization potency of the trimeric TriSb92 inhibitor may also to stem from its mode of action. Instead of competing with the high-affinity interaction between ACE2 and the RBD of spike, TriSb92 prevents SARS-CoV-2 entry by triggering a series of conformational changes that initially involve the positioning of the RBD subunits of the spike trimer. which appear to trigger additional changes involving the S2 region and loss of a functional prefusion conformation More specifically, our cryo-EM results indicate that engagement with TriSb92 induces an all three RBDs up state of spike, as evidenced by the high prevalence of the all up RBD conformation, which is known to be rare except when induced by a binding partner[29–32]. Interestingly, the binding site of TriSb92 is inaccessible in the down-conformation of the RBD. We postulate that once

an RBD of the spike samples the up-conformation (the common one RBD up state), a subunit of TriSb92 binds to the RBD, and the remaining two free TriSb92 subunits readily sequester ensuing up-conformations of the two other RBDs until an all-up conformation is achieved. This model is consistent with our findings that following TriSb92 treatment, the S-trimer presents primarily in an all-up state (~54%) fully engaged with the sherpabody, with smaller populations of mixed up/down and all-down states (Figs. S5, S6). However, the all three RBDs up state observed in our cryoEM analysis, performed using a prefusion-stabilized (K986P/V987P mutations) spike ectodomain with an abrogated furin cleavage site and a C-terminal trimerization domain, appears to reflect only the initial stage of this inhibitory conformational change, which in the case of native spike trimer on the viral membrane results in a more fundamental structural reorganization. Treatment of spike-decorated pseudoviral particles with TriSb92 induced an altered conformation in the S2 subunit of spike that was evidenced by a failure to detect cleaved S2 or S2-containing S0 precursor protein by Western blotting. Instead, the amount of virion-associated cleaved S1 protein remained unchanged, indicating that TriSb92 treatment did not lead to shedding of the S1 subunit, as has been demonstrated for certain antibodies that target RBD epitopes overlapping with the TriSb92 binding region[33,34]. In agreement, we also did not observe any decrease in virion attachment to ACE2 receptor-expressing target cells by TriSb92. Thus, we conclude that the inhibitory mechanism of TriSb92 involves pushing of virion-associated spike from its native prefusion state into a bona fide post-fusion or another conformation that cannot support productive viral entry after receptor binding.

TriSb92 was built from a sherpabody clone (Sb92) derived from phage-library screening without any further affinity optimization. Thus, it should be relatively facile to achieve an even higher inhibitory potency by affinity maturation of Sb92 and perhaps also by optimization of the trimerization linkers in TriSb92. However, this may not be of great practical relevance considering the already low $IC_{50}$ concentrations of TriSb92 and the ease of producing it in very high quantities in low-cost bacterial expression systems. Moreover, the extended on-shelf stability in ambient temperatures render TriSb92 a suitable candidate for nasal spray products.

Treatment of mice intranasally with as little as 5 ug of TriSb92 could effectively protect them from a subsequent challenge with SARS-CoV-2. The intranasal dose of TriSb92 that should be administered to susceptible persons at risk for SARS-CoV-2 exposure remains to be established but is likely to be substantially lower than extrapolation of our current data on mice that were challenged by inoculation of a rather massive amount of SARS-CoV-2 ($2 \times 10^5$ TCID of B.1.351 or $5 \times 10^6$ of maVie16) into their respiratory tract. Likewise, compared to the current mouse model data the time window allowed for administering TriSb92 before or after natural SARS-CoV-2 exposure to prevent active viral replication and disease could be wider. Moreover, even in cases when TriSb92 would fail to prevent the infection from progressing, by limiting the early amplification of the virus in the epithelium of the nasal cavity and nasopharynx and thereby delaying the infectious process TriSb92 could effectively tip the balance in favour of the host and thereby help to reduce the risk of severe COVID-19 disease.

The pressure on healthcare systems posed by the continuously emerging viral variants emphasizes the need for new innovations to tackle the rapidly evolving immune escape potential of SARS-CoV-2, and to prepare for future pandemics caused by related sarbecoviruses evolving in the nature. Our current data show that TriSb92 represents a valuable addition to the arsenal of available measures for addressing these challenges.

## Methods
Research reported in this study complies with all relevant ethical regulations, and our animal study protocol has been approved by the

Animal Experimental Board of Finland (license number ESAVI/ 28687/2020).

### Phage panning
Sherpabody phage display library (size ~$10^{11}$ cfu) was obtained from Next Biomed Therapies Oy. To develop sherpabodies specific for SARS-CoV-2 spike receptor binding domain (RBD), phage affinity selection process was conducted using standard solid phase sorting strategy. Specific phage-displayed sherpabodies were selected for by panning against RBD-mouse IgG2a Fc-fusion protein[35]. Three sequential rounds of off-target depletion and specific panning were performed using a control mFc and specific RBD-mFc, respectively. The immobilized control and target proteins (30 µg/ml in PBS; Maxisorp Immunotubes, Nunc) were sequentially incubated in the presence of infectious naïve sherpabody phage library in 2.5% milk-PBS-0.1% Tween20, for 2 h (RT) and o/n (4 °C), respectively. Non-specific phages were removed by extensive washing (PBS-0.05% PBS-Tween), and the remaining pool of phage were eluted and amplified in E. coli XL1-Blue host cells (Avantor; #AGLS200249) according to standard protocols. The amplified pool of phages was collected, and the process was reiterated over three rounds to enrich phage-displayed sherpabodies specific to the RBD-target protein.

### Screening and characterization of RBD-binding clones by ELISA
Representative transformants from the second and third panning rounds were tested for specific binding to the RBD-mFc target protein by enzyme-linked immunosorbent assay (ELISA). Mimicking the affinity selection process, in phage-ELISA the target and control proteins were immobilized on an immunoplate (Maxisorp, Nunc). Each single colony represents progeny from a single E. coli cell that harbours a phagemid expressing a unique sherpabody-pIII fusion protein. Specifically, phage-ELISA was performed in 96-well Maxisorp microtiter plates (Nunc) coated over night at 4 °C with 100 µl (10 µg/ml in PBS) of target (RBD-mFc) and control proteins (Anti-E-tag antibody; GE Healthcare; cat# 27941201 V; lot# 355351). The wells were washed 3 x with PBS-0.05% Tween20 and blocked with 5% skimmed milk powder in PBS (milk-PBS) for 2 h at RT. Appropriate dilutions of sherpabody-displaying single phage clones were prepared in milk-PBS and incubated with the coated target protein for 1 h at RT followed by washes 3 x with PBS-0.05% Tween20 to remove unbound phage. The detection was performed with HRP-conjugated mouse monoclonal anti-M13 antibody (1:5000; GE Healthcare/Cytiva; Cat# RPN1236; Lot# 385982)), and TMB (3,3′ 5,5′-tetramethylbenzidine) substrate (Thermo Fisher; Cat#34028). The staining reaction was stopped with 1 M sulfuric acid and absorbance measured at 450 nm using Hidex Sense Microplate Reader. The DNA encapsulated by the positive phage clones was then sequenced and translated to determine the sequence of the displayed sherpabody.

### Sherpabody production and purification
The discovered unique sherpabodies were cloned and expressed in E. coli as monomeric and/or trimeric GST-fusion proteins using standard protocols (GE Healthcare). 15 mer Gly-Ser, $(GGGGS)_3$, linkers were applied in between each fusion partner. GST-tagged sherpabodies were purified by affinity chromatography using glutathione sepharose according to manufacturer's instructions (GE Healthcare). The buffer was exchanged to PBS by dialysis (140 mM NaCl, 10 mM phosphate buffer, 3 mM KCl, pH 7.4), and the purified sherpabodies were stored at −80 °C, −20 °C and/or at + 4 °C.

### ELISA
Affinity analysis by semi-quantitative sandwich-ELISA was performed using plastic-coated (o/n, 4 °C) GST-sherpabody fusion protein followed by binding of a concentration series of soluble RBD-His target protein[35] (1 h, RT). The washes and detection was performed as

described above using monoclonal HRP-conjugated anti-His antibody (1:5000; Invitrogen; Cat# MA1-21315-HRP; Lot# 2312647).

GST-Sb92/RBD-His/ACE2-mFc sandwich-ELISAs were performed in two ways: first using plastic-coated GST-Sb92 as the capturing reagent (10 µg/ml) and ACE2-mIgG2a as the detection reagent (10 µg/ml) and then vice-versa (10 µg/ml and 5 µg/ml, respectively) to analyse their ability to simultaneously bind to RBD-His[35] (2-fold dilution series starting from 10 µg/ml). The complex formation was detected using either HRP-conjugated goat anti-Mouse-IgG (Sigma Aldrich; 1:5000) or mouse monoclonal anti-GST (1:5000; GE Healthcare; Lot# 17007382)) antibodies. Her2-binding GST-Sb1206 served as a negative control.

The specificity (off-target) of TriSb92 binding to RBD was investigated by ELISA. Whole cell lysates of primary human nasal epithelial cells (HNECs; see isolation protocol below) were prepared by lysing $10^6$ cells/ml in ice cold Lysis buffer (50 mM Tris pH 7.4, 150 mM NaCl, 1% NP-40, 1 mM EDTA) for 30 min, on ice (vortexing every 5–10 min). Nunc Maxisorp 96-well plates were coated with the concentrated cell lysates o/n, + 4 °C (100 µl/well, 1:100 in PBS). Whole cell lysate derived from HNECs transduced with an adenoviral vector directing the expression of cell-surface expressed RBD of SARS-CoV-2 spike protein were used as a positive control. The ELISA was performed similarly as described above using standard protocols.

## Neutralization assays

HEK293T (ATCC; Cat# CRL-3216) and HEK293T-ACE2 cells were maintained in DMEM supplemented with 10% fetal bovine serum, 2% L-Glutamine, and 1% penicillin/streptomycin (complete medium). Angiotensin-converting enzyme 2 (ACE2) expressing HEK293T cells were generated by lentivirus-mediated gene transduction. Briefly, pWPI-puro plasmid containing ACE2 cDNA (AB046569.1) was co-transfected with p8.9NdSB and vesicular stomatitis virus G protein (VSV-G) expressing envelope plasmids into HEK293T cells in complete medium using polyethylenimine. The recombinant lentivirus containing supernatant was collected 48 h post-transfection, filtered and used to infect wild-type HEK293T cells. Transduced cells were selected with puromycin.

Luciferase encoding SARS-CoV-2 pseudotyped reporter virus was generated by transfecting HEK293T cells with p8.9NdSB, pWPI-GFP expressing Renilla luciferase, and pCAGGS, an expression vector containing the SARS-CoV-2 S protein cDNA of the Wuhan-Hu-1 reference strain (NC045512.2). The last 18 amino acids containing an endoplasmic reticulum (ER)-retention signal of the spike protein was removed to enhance transport to the plasma membrane. SARS-CoV-1 pseudovirus was similarly constructed. Pseudovirus stocks were harvested 48 h after transfection, filtered and stored at −80 °C. Inserts for the pseudotyping expression vectors producing spike glycoproteins encoded by the original Wuhan-Hu-1 strain (with D614G mutation) or its variants B.1.351 (Beta), B.1.617.2 (Delta), and B.1.1.529 (Omicron), including the subvariants BA.1, BA.3, BA.2, BA.2.13, BA.4/5, BA.2.3.20, BF.7, BQ.1.1, XBB, BU.1, or SARS-CoV-1 were assembled from synthetic DNA fragments (Integrated DNA Technologies) using Gibson assembly cloning kit (New England Biolabs; Cat1 E5510S) and verified by DNA sequencing.

To assess its neutralizing activity TriSb92 was serially diluted in complete medium. 12.5 µl of TriSb92 dilutions were mixed with 37.5 µl of luciferase encoding SARS-CoV-2 pseudotyped reporter viruses in 96-well cell culture plates and incubated at 37 °C for 30 min. After incubation, 20,000 HEK293T-ACE2 cells (in 50 µl) were added to the wells and the plates were further incubated at 37 °C for 48 h. The amount of internalized pseudovirus in infected cells was quantified by measuring luciferase activity using Renilla-GLO assay (Promega; Cat6 E2710). The relative luciferase units were normalized to those of control samples. The pseudovirus neutralization experiments were repeated independently three or more times with similar results. The

average and standard deviations of representative assays performed in duplicates is shown.

To assess the inhibitory activity of TriSb92 against clinical virus isolates, the neutralization was performed as described above for the pseudoviruses with the following modifications: 40 000 VeroE6-TMPRSS2-H10 cells[36] were seeded per well and mixed with virus-inhibitor mixture, which was replaced with complete DMEM 1 h post infection (p.i) at 37 °C. Nine hours later the cells were fixed with 4% paraformaldehyde, permeabilized with 3% BSA-PBS-0.3 % Triton-X100, and labelled with rabbit polyclonal anti-SARS-CoV-2 nucleoprotein (NP; 1:1000; Rockland Immunochemicals; Cat3 200-401-A50; Lot7 200-402-A50) primary antibody and goat Alexa Fluor 488-conjugated (Invitrogen; 1:5000) anti-rabbit polyclonal secondary antibody. The cell nuclei were visualized by Hoechst staining, and the percentage of cells that labelled positive for SARS-CoV-2 NP was quantified using Opera Phenix High Content Screening System (PerkinElmer). Each experimental condition involved five replicate wells, and an average of 15,000 cells (total > 70,00 per condition) per well were analyzed. Neutralization of authentic viruses were repeated with similar results three times for Wuhan, twice for Delta, BA.1, and BA.5 and once for Beta in experiments performed in duplicate and involved counting of approximately 25,000 cells per condition after immunostaining illustrated in the figure insert. The original data are provided as a Source Data file

The clinical isolates used in this study were isolated in Finland from patient nasopharyngeal samples. Beta variant is the isolate hCoV-19/Finland/THL-202101018/2021 (EPIISL_3471851), Delta is hCoV-19/Finland/THL-202117309/2021 (EPI_ISL_2557176), Omicron/BA.1 is hCoV-19/Finland/THL-202126660/2021 (pending for the Gisaid ID), Omicron/BA5.1 is hCoV-19/Finland/THL-202213593/2022 (EPI_ISL_13118918, GeneBank ID OP435368). Viruses were propagated in VeroE6-TMPRSS2-H10 cells[36]. The generation of mouse-adapted maVie16 SARS-CoV-2 is described elsewhere[17]. All cell lines used had a well-traceable origin and were authenticated in our laboratory by light microscopic morphological analysis only. All cells lines have been tested negative for mycoplasma contamination. No commonly misidentified cells were used in the study.

## Cell viability assay

Primary human nasal epithelial cells (HNECs) were isolated by brushing from the nasal cavity of a male donor after informed consent. The isolated cells were cultured and maintained in PneumaCult™-Ex Plus Basal Medium supplemented with 1x PneumaCult™-Ex Plus Supplement, 1x hydrocortisone (200x stock solution 96 µg/ml) and 1x penicillin-streptomycin (all from StemCell Technologies). For cell expansion, the culture flasks and plates were coated with Collagen Solution according to manufacturer's instructions, and Animal Component-Free Cell Dissociation Kit was used for dissociation and passaging (all from StemCell Technologies).

For the cell viability assay, HNECs were cultured on an opaque-walled 96-well plate (10 000 cells/well) for 24 h, and then treated with a 2-fold concentration series of TriSb92 diluted in complete culture medium (starting from 1 mg/ml) for 24 h at 37 °C. $NaN_3$ (5%) served as a cytotoxic drug control. Cell viability was measured using CellTiter-Glo 2.0 Assay (Promega; Cat# E2710) according to manufacturer's instructions. Average and standard deviations of a representative assay performed in triplicates were calculated.

## Stability of TriSb92 during long-term on-shelf storage

The stability of TriSb92 after long-term storage in different temperatures was investigated using pseudovirus neutralization assay using Wuhan-Hu-1 reference strain. TriSb92 was stored as 1 mg/ml stocks in PBS (140 mM NaCl, 10 mM phosphate buffer, 3 mM KCl, pH 7.4) in room temperature [RT; + 21–25 °C], fridge [+6–8 °C] or freezer [−20 °C or −80 °C] for 1–15 months (mo).

## Animal studies

Balb/c mice (Envigo) were transported to the University of Helsinki (Finland) biosafety level 3 (BSL-3) facility and acclimatized to individually ventilated biocontainment cages (ISOcage; Scanbur) for seven days with ad libitum water and food (rodent pellets). The mice were kept 4–6 h in the light during the day time, and the rest of the day i.e., 18–20 h in the dark. The temperature and humidity in the BSL-3 laboratory and cages was kept between 21–23 °C and 25–35%, respectively. After the acclimatization period, 9-week old female Balb/c were placed under isoflurane anesthesia and intranasally inoculated with 25 µl per nostril of TriSb92 (25 or 2.5 µg/nostril). The mice were challenged by infection with 20 µl of SARS-CoV-2 B.1.351 clinical isolate ($2 \times 10^5$ PFU) or with mouse-adapted maVie16 SARS-CoV-2[17] ($0.5–1 \times 10^6$ PFU). Immediately following the inoculation, the isoflurane was switched off and the animals were held in an upright position for a few seconds to allow the liquid to flush downwards in the nasal cavity. All mice were weighed on a daily basis, and their wellbeing was carefully monitored throughout the experiment for signs of illness (changes in posture or behaviour, rough coat, apathy, ataxia and weight loss), but none of the mice showed any clinical signs. Euthanasia was performed 2 days post infection under terminal isoflurane anesthesia with cervical dislocation. All animals were dissected immediately after death and the right lungs collected for virological examination. The left lung, remaining thoracic organs and heads (animals from the first experiment) were fixed in 10% buffered formalin for 48 h and stored in 70% ethanol for histological and immunohistochemical examinations.

## RNA isolation and RT-PCR

RNA was extracted from the right lung of the mice using Trizol (Thermo Scientific) according to the manufacturer's instructions. Isolated RNA was directly subjected to one-step RT-qPCR analysis using TaqMan fast virus 1-step master mix (Thermo Scientific) and AriaMx instrumentation (Agilent) as described previously for E and subE genes[37]. The RT-qPCR for actin was conducted as previously described in ref. [38]. Relative quantification of actin-normalized viral RNA levels was achieved by the comparative Ct method[39] using the average of non-treated animals as reference.

## Histology and immunohistochemistry

The left lung, remaining thoracic organs and heads were trimmed for histological examination. Heads were sawn longitudinally in the midline using a diamond saw (Exakt 300; Exakt) and gently decalcified in RDF (Biosystems) for 5 days at room temperature and on a shaker. Tissues were routinely paraffin wax embedded. Consecutive sections (3–5 µm) were prepared from lungs and heads and routinely stained with hematoxylin-eosin (HE) or subjected to immunohistochemistry for the detection of SARS-CoV-2 antigen, as previously described in ref. [16].

## Recombinant SARS-CoV-2 S trimer production and complex formation

To express the SARS-CoV-2 S, a gene encoding for the prefusion stabilized S protein ectodomain[19] was produced as synthetic cDNA (GeneArt, Life Technologies). The cDNA template encoded for the residues 14 – 1208 of the original Wuhan-Hu-1 strain S protein (NCBI Reference Sequence: YP_009724390.1) with prefusion-stabilizing proline substitutions at residues 986 and 987, an abrogated furin S1/S2 cleavage site with a GSAS substitution at residues 682–685, and a C-terminal T4 fibritin trimerization motif followed by an HRV3C protease cleavage site, SpyTag003, and 8xHisTag. The gene was cloned into the mammalian expression vector pHLsec (Adgene) and transfected into Expi293F™ (Thermo Fisher Scientific) suspension cells at a density of $3 \times 10^6$ cells per ml using the ExpiFectamine™ 293 Transfection Kit (Thermo Fisher Scientific). Following 6 days of cultivation on an orbital shaker a 36.5 °C and 5% CO$_2$, the S protein containing

supernatant was collected, clarified by centrifugation, and filtered through a 0.45 µM filter. Imidazole was added to the supernatant to 3 mM final concentration, and SARS-CoV-2 S protein was purified from the supernatant by immobilized nickel affinity chromatography with a 1 ml HisTrap excel column (Cytiva) using 300 mM imidazole for elution. S-protein containing eluate was concentrated and buffer exchanged to 10 mM Tris pH 8 + 150 mM NaCl buffer using an Amicon Ultra centrifugal filter (MWCO 100 kDa, Millipore). Prior to grid preparation, pure TriSb92 was added to a purified S-trimer aliquot at 1.5x molar excess, and the complex was incubated on ice for 15 min.

## Cryo-EM grid preparation, data acquisition and data processing

A 3 µl aliquot of a pure, prefusion SARS-CoV-2 S-trimer (0.3 mg/ml) mixed with TriSb92 (0.05 mg/ml) was applied on Quantifoil 1.2/1.3 grids (1.2 µm hole diameter, 200 mesh copper) that had been glow discharged in a plasma cleaner (PDC-002-CE, Harrick Plasma) for 30 s. The grids were blotted for 8 s and plunged into liquid ethane using a vitrification apparatus (Vitrobot, Thermo Fisher Scientific). Data were collected on a Titan Krios transmission electron microscope (Thermo Fisher Scientific) equipped with Gatan K2 direct electron detector using electron exposure of 55 e⁻/Å² per image at a nominal magnification of 165,000x, resulting in a pixel size of 0.82 Å (Table S1). Data were processed in cryoSPARC[40] (Fig. S5). Contrast transfer function parameters were estimated by CTFFIND4[41]. Particles were picked with Topaz[42], using a subset of S-trimer particles first pruned by 2D classification as the training set. After another round of 2D classification to discard false positive particles, an initial volume of the S trimer was calculated ab initio. After 3D refinement of particle poses and shifts, local motion correction, extraction of particles with box size of 512x512 and second round of refinement with C3 symmetry, the symmetry was expanded, and the particles were subjected to 3D variability analysis[43] which focused to the RBD region. This revealed three main classes of particles: all RBDs up, all RBDs down, and a mixed up and down population (Fig. S5, Fig. S6). Maps of the all up and all down classes were reconstructed with non-uniform refinement. A map for all RBDs down (reconstruction B) class was calculated applying C3 symmetry, after removing the symmetry copies created by symmetry expansion. For all RBDs up map, no symmetry was applied (reconstruction A). Instead, the RBD region density was further refined by local, focused refinement using particles where the bulk of the S-trimer density had been subtracted. The resulting maps were filtered to local resolution for further analysis. CryoEM data collection and processing statistics are presented in Table S1.

## Fitting SARS-CoV-2 S and Sb92 into the cryo-EM reconstructions

A molecular model of the SARS-CoV-2 S-trimer, derived from PDB 7KMS[44], was docked into our reconstruction A of the SARS-CoV-2 spike using the fitmap function in UCSF ChimeraX[45]. In short, a map was simulated for the S-trimer structure to the resolution of 2.9 Å, to match the resolution of the spike reconstruction, and fitting was performed using fitmap global search. The top solution yielded a map-to-map correlation score of 0.78, confirming that, similar to structure 7KMS, our reconstruction of the S-trimer displays a 3-up RBD conformation. We observed that the innate flexibility of RBDs resulted in weaker density features in RBD regions of the full spike reconstruction, and so, RBDs were modelled into locally refined RBD-region density (Figs. S5, S8). First, an RBD was separated from the 7KMS spike structure in Coot[46] and fitted into the locally-refined RBD region reconstruction using fitmap in ChimeraX, yielding a correlation score of 0.86. Furthermore, additional density not comprised of the S-protein was observed adjacent to each RBD (Fig. 6). To confirm whether this density was comprised of Sb92, the cryoEM reconstruction was segmented around the RBD using color zone (coloring radius of 2.5 Å) and split map functions in Chimera, followed by fitting a model of Sb92, predicted by I-TASSER[21] to high confidence (C-score 0.51) and geometry

optimized in Phenix[47], into the additional density with a fitmap global search. The top solution placed Sb92 into the density adjacent to RBDs with a correlation score of 0.78. Furthermore, this sherpabody placement reveals that the RT-loop, which contains antigen-binging mutations in Sb92 (Fig. 1), forms a key interface with the RBD (Fig. S8). Residues comprising the RT-loop were refined into the density using ISOLDE[48] within ChimeraX. Finally, three copies of the RBD−Sb complex were fitted back into the reconstruction of the full spike (reconstruction A) and connected to the rest of the fitted spike derived from 7KMS in Coot, completing the model of the SARS-CoV-2 spike in complex with Sb92.

Our second reconstruction B (Fig. S7), was validated as a closed conformation by fitting a SARS-CoV-2 structure (PDB 6ZP0[49], presenting a 3-down RBD conformation) into the density. Fitmap global search was performed between a map simulated for structure 6ZP0 at resolution of 3.1 Å, and our cryoEM reconstruction B, yielding a map-to-map correlation score of 0.76 and leaving no unexplained density.

## Molecular graphics and protein interface analysis

Molecular graphics images were generated using PyMOL (The PyMOL Molecular Graphics System, Version 2.5.0, Schrödinger, LLC) and C[5151]himeraX v1.5, developed by the Resource for Biocomputing, Visualization, and Informatics at the University of California[45]. Residues comprising the interface of SARS-CoV-2 S with ACE2 receptor and inhibitors, including TriSb92 and therapeutic monoclonal antibodies, were identified using the PDB ePISA server.

## Reporting summary

Further information on research design is available in the Nature Portfolio Reporting Summary linked to this article.

## Data availability

The cryo-EM density maps of i) SARS-CoV-2 spike with RBDs in the "up"-conformation and decorated with TriSb92 (accession code EMD-16383) and ii) SARS-CoV-2 spike with RBDs in the "down" conformation (EMD-16388) have been deposited in the EMDB at the EBI. Coordinates of the fitted SARS-CoV-2 spike in complex with Sb92 have deposited in the PDB (accession code PDB 8C1V). All data generated or analysed during this study are included in the main text and its supplementary information files. Other structures used in this study were obtained from the PDB with accession codes 1S1N (nephrocystin SH3 domain), 7KMS (SARS-CoV-2 S-trimer with all RBDs in the up conformation bound to ACE2), 6ZP0 (SARS-CoV-2 S-trimer with all RBDs in the down conformation), 7A29 (sybody-bound SARS-CoV-2 spike), and 6W41, 7KMG, 7C01, 6XDG RBD-Fab complexes. Source data are provided with this paper.

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

## Acknowledgements

We thank Virpi Syvälahti for expert technical assistance, Mikael Ritvos for help with recombinant protein production, Giuseppe Balistreri and Ravi Kant Oiha for human nasal epithelial cells and related advice, Michael Hall, Benita Löflund, and Pasi Laurinmäki for support with electron microscopy. The facilities and expertise of the HiLIFE CryoEM unit at the University of Helsinki, a member of Instruct-ERIC Centre Finland, FINStruct, and Biocenter Finland are gratefully acknowledged. The data was collected at the Umeå Core Facility for Electron Microscopy, a node of the Cryo-EM Swedish National Facility, funded by the Knut and Alice Wallenberg, Family Erling Persson and Kempe Foundations, SciLifeLab, Stockholm University and Umeå University. We are grateful for the facilities and expertise provided to us by the FIMM Genomic Center at the University of Helsinki, CSC - IT Center for Science, Finland, and the Histology Laboratory of Institute of Veterinary Pathology at the University of Zurich. This work was supported by the Academy of Finland (grants 336492 to J.T.H., 342988 to I.R., 336490 and 339510 to O.V., 336425 and 331787 to K.S.), E.U. Horizon 2020 programme (grant 874735 to O.V.) as well as funding from the Jane and Aatos Erkko Foundation (O.V.) and the Sakari Alhopuro Foundation (K.S.).

## Author contributions

A.R.M. and K.S. designed the study and analysed the data. A.R.M., H.U., P.S., J.H., L.L., R.F., and K.S. were responsible for the inhibitor development, characterization, and validation. A.P., R.N., and O.R. produced and purified the recombinant proteins. S.M., A.H., and P.O. performed the studies with the clinical virus isolates. R.K., L.K., T.St., S.K., T.Si., and O.V. were involved in the infectious mouse model and animal studies. A.K. conducted mouse pathology and immunohistochemistry. Structural biology studies were designed, conducted, and analyzed by I.R., L.H., and J.T.H.

## Competing interests

K.S. is a founder and shareholder of Next Biomed Therapies Oy that develops SH3 scaffold targeting technologies. ARM is a founder and shareholder of the start-up Pandemblock Oy that has acquired commercial rights for TriSb92. The authors are inventors of the related patents/applications WO2017009533 (K.S.) and PCT/FI2022/050764 (K.S. and A.R.M.). The other authors have no competing interests to declare.
