## [Peer Review File · Nature Communications]

Intranasal trimeric sherpabody inhibits SARS-CoV-2 including recent immunoevasive Omicron subvariantsEditorial Note: This manuscript has been previously reviewed at another journal that is not operating a transparent peer review scheme. This document only contains reviewer comments and rebuttal letters for versions considered at Nature Communications.

Reviewers' Comments:

Reviewer #1:

Remarks to the Author:

The authors have sufficiently addressed all of my concerns. The addition of data from a lethal animal model is very powerful and really improves the impact of this manuscript. Overall this is convincing work. I only have a few minor comments.

1. Line 115: Please don't use the term "very pleasing" to describe a scientific result.
2. Line 142: I'm not sure you should say "insensitive" considering that there are some modest changes to the IC50 values in different isolates and between SARS-CoV-2 and SARS-CoV-1.
3. Line 322/341: The term "highly conserved" is relative to what is being evaluated. Is the region highly conserved amongst all CoVs or just among the Sarbecoviruses? Make sure to make this clear. If there is significant conservation amongst other CoVs the authors should discuss the ability of this antibody to perhaps target other more divergent CoVs.
4. In the discussion the authors state that "TriSb91 prevents SARS-CoV-2 entry by triggering a conformation change in the spike trimer". I'm not sure the data unequivocally indicates that this change is what leads to inhibition of viral entry. The authors even describe other changes that occur that impact the processing of the spike protein. Do the other changes first require the change to the spike trimer? If so perhaps this needs to be better explained. Otherwise its not clear exactly which conformational change truly leads to a reduction in viral entry.

Reviewer #2:

Remarks to the Author:

The authors addressed all my concerns. The science is greatly improved and I support the publication of this manuscript. Congratulations!

Reviewer #4:

Remarks to the Author:

The cryo-EM structures and analysis were sufficiently improved.

Parts of the cryo-EM results are still not faithfully described by the text and figures. Specifically -

Lines 295-299 The authors write: "Single-particle cryoEM analysis of the prefusion SARS-CoV-2 S trimer mixed with TriSb92 resulted in two reconstructions with average estimated resolutions of 2.9 Å and 3.1 Å (Fig. S5, Fig. S6 and Table S1; 6–7Å resolution over the flexible RBD regions) that display the distinct "up" and "down" conformations characteristic of sarbecoviral RBDs." - According to Fig S6 this is not true. The data comprises many conformations – all-up (54%), all-down (~11%) and mixed up and down (35%). The all-up is obviously the important species, but the all-down is a minority and

was chosen simply because it has C3 symmetry, and is therefore easier to reconstruct to high resolution. As presented now in the text and Fig S5 it looks as if the complex has 2 distinct conformations. Can the authors please make this clear in the text and figures. A more accurate representation of the data is presented in the Letter Figure 1. Such a figure should appear in the manuscript.

Please add angular distribution and 3D variability results (significant modes and component plots) to the supplementary figures. These are important results.

Line 373-374 The authors write: "Our cryo-EM results show that engagement with TriSb92 induces an "all three RBDs up" state of spike." - This is not supported by the EM data. Such a statement requires comparison to a negative control (same construct without the SHERPabody), which is not present here. Perhaps a comparison to other papers will show that the percentage of open conformations is raised by the addition of TriSb92.

Reviewer #1:

1. Line 115: Please don't use the term "very pleasing" to describe a scientific result.

We have now rephrased the sentence and replaced "very pleasing" with a more neutral and specific term "subnanomolar".

2. Line 142: I'm not sure you should say "insensitive" considering that there are some modest changes to the IC50 values in different isolates and between SARS-CoV-2 and SARS-CoV-1.

We have now rephrased the sentence, and replaced "insensitive" with "not prevented by".

3. Line 322/341: The term "highly conserved" is relative to what is being evaluated. Is the region highly conserved amongst all CoVs or just among the Sarbecoviruses? Make sure to make this clear. If there is significant conservation amongst other CoVs the authors should discuss the ability of this antibody to perhaps target other more divergent CoVs.

We have now specified the meaning: "site that is highly conserved among sarbecoviruses"

4. In the discussion the authors state that "TriSb91 prevents SARS-CoV-2 entry by triggering a conformation change in the spike trimer". I'm not sure the data unequivocally indicates that this change is what leads to inhibition of viral entry. The authors even describe other changes that occur that impact the processing of the spike protein. Do the other changes first require the change to the spike trimer? If so perhaps this needs to be better explained. Otherwise its not clear exactly which conformational change truly leads to a reduction in viral entry.

This has now been better explained.

Reviewer #2

No further questions

Reviewer #4

1. Parts of the cryo-EM results are still not faithfully described by the text Specifically -

1(a). Lines 295-299 The authors write: "Single-particle cryoEM analysis of the prefusion SARS-CoV-2 S trimer mixed with TriSb92 resulted in two reconstructions with average estimated resolutions

of 2.9 Å and 3.1 Å (Fig. S5, Fig. S6 and Table S1; 6–7Å resolution over the flexible RBD regions) that display the distinct “up” and “down” conformations characteristic of sarbecoviral RBDs.” - According to Fig S6 this is not true. The data comprises many conformations – all-up (54%), all-down (~11%) and mixed up and down (35%). The all-up is obviously the important species, but the all-down is a minority and was chosen simply because it has C3 symmetry, and is therefore easier to reconstruct to high resolution. As presented now in the text and Fig S5 it looks as if the complex has 2 distinct conformations. Can the authors please make this clear in the text and figures. A more accurate representation of the data is presented in the Letter Figure 1. Such a figure should appear in the manuscript.

As suggested, a new figure similar to that included in our earlier rebuttal letter presenting the observed spike populations has now been added (Supplementary Figure 6). The presence of mixed Spike populations in addition to “all-up” and “all-down” has now also been clearly discussed.

1(b). Please add angular distribution and 3D variability results (significant modes and component plots) to the supplementary figures. These are important results.

Angular distribution plots have now been added to the revised Supplementary Figure S7.

1(c). Line 373-374 The authors write: “Our cryo-EM results show that engagement with TriSb92 induces an “all three RBDs up” state of spike.” - This is not supported by the EM data. Such a statement requires comparison to a negative control (same construct without the Sherpabody), which is not present here. Perhaps a comparison to other papers will show that the percentage of open conformations is raised by the addition of TriSb92.

Reviewer #4 is correct in pointing out that the rarity of the “all three RBDs up” state of spike in the absence of binding to any inhibitor or neutralizing antibody has been well established in the literature, but was not directly addressed in our study. Citations to four earlier studies demonstrating this (*Barnes, et al., Cell 182, 828-842, 2020; Ke et al., Nature 588, 498-502, 2020; Yan et al., Cell Res 31, 717-719, 2021; Lu et al., Cell Host Microbe 28, 880-891, 2020*) have now been included to back up this notion in our Discussion.